# SoloParkour: Constrained Reinforcement Learning for Visual Locomotion from Privileged Experience

**Elliot Chane-Sane**[*1], **Joseph Amigo**[*12], **Thomas Flayols**[1],
**Ludovic Righetti**[23], **Nicolas Mansard**[13]
[1]LAAS-CNRS, Université de Toulouse, CNRS, Toulouse, France
[2]Machines in Motion Laboratory, New York University, USA
[3]Artificial and Natural Intelligence Toulouse Institute, Toulouse, France
https://gepetto.github.io/SoloParkour/

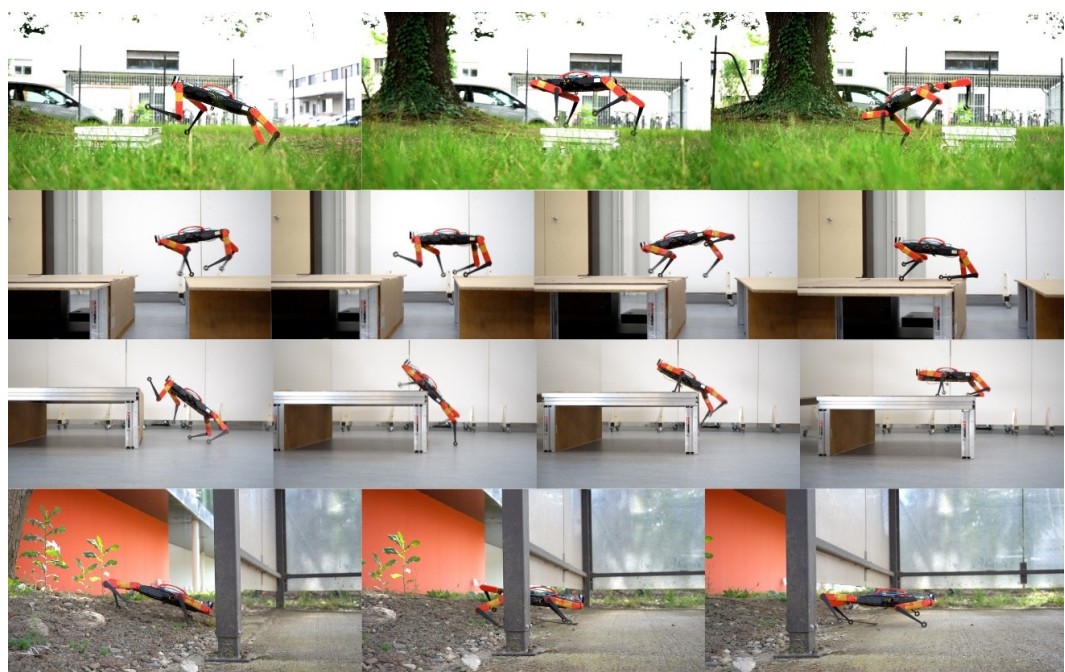

Figure 1: The open-hardware quadruped robot Solo-12 performs agile skills that are reminiscent of parkour, such as walking, climbing high steps, leaping over gaps, and crawling under obstacles.

**Abstract:** Parkour poses a significant challenge for legged robots, requiring navigation through complex environments with agility and precision based on limited sensory inputs. In this work, we introduce a novel method for training end-to-end visual policies, from depth pixels to robot control commands, to achieve agile and safe quadruped locomotion. We formulate robot parkour as a constrained reinforcement learning (RL) problem designed to maximize the emergence of agile skills within the robot's physical limits while ensuring safety. We first train a policy without vision using privileged information about the robot's surroundings. We then generate experience from this privileged policy to warm-start a sample efficient off-policy RL algorithm from depth images. This allows the robot to adapt behaviors from this privileged experience to visual locomotion while circumventing the high computational costs of RL directly from pixels. We demonstrate the effectiveness of our method on a real Solo-12 robot, showcasing its capability to perform a variety of parkour skills such as walking, climbing, leaping, and crawling.

**Keywords:** Reinforcement Learning, Agile Locomotion, Visuomotor Control

---

[*]Equal contribution. Correspondence to elliot.chane-sane@laas.fr

8th Conference on Robot Learning (CoRL 2024), Munich, Germany.

# 1 Introduction

In recent years, training locomotion policies in simulation using reinforcement learning (RL) then transferring them to real robots has proven highly effective in mastering agile skills [1, 2, 3, 4, 5, 6, 7, 8, 9, 10]. Notably, [11, 12, 13, 14] have successfully demonstrated a range of dynamic skills akin to the athletic maneuvers seen in parkour, including walking, climbing, jumping, and crawling, which necessitates tight coordination between vision and control. In this work, we are interested in developing parkour skills from onboard depth inputs for Solo-12, a lightweight, open-hardware quadruped robot [15], using this sim-to-real approach. Solo is a lighter prototype than other quadruped robots experimented in parkour projects (e.g. Atlas, Go-1, or Anymal). Safe visual locomotion with Solo is then at its highest stake, to prevent exceeding the limited torque range of its motors and breaking the 3D-printed plastic shells and exposed electronic components in unexpected impacts or falls.

Yet, training visual locomotion policies presents significant challenges due to the limited field of view and dynamic, lag-prone visual perception during dynamic movements. Previous methods for end-to-end control from pixels [16, 12, 13] typically involve a two-stage process, where a policy is first trained with privileged information about the robot's surroundings, then the learned behaviors are distilled into a visual policy using imitation learning [17]. However, such an approach relies on the unrealistic assumption that privileged information can be fully reconstructed from a history of depth images. Indeed, such privileges may reveal information obscured behind obstacles, outside the field of view of the egocentric camera, or simply missing from the depth image stream due to lags. This gap between the privileged information and the actual visual inputs may result in behaviors that a vision-based policy cannot replicate accurately. In that case, an optimal vision policy would differ from the one using privileged information. It would, for example, try to gather more information before crossing an obstacle to handle the occlusion. Ideally, training RL directly from pixels would ensure the policy learns behaviors effectively based on its actual visual sensors, but this method is impractical due to the high computational costs associated with generating depth images in simulations [16].

To address these challenges, we propose *SoloParkour*, a novel approach for training parkour locomotion policies using depth inputs. Our method frames parkour as a constrained reinforcement learning problem [18, 19, 20]. This framework allows the RL agent to explore the full capabilities of the robot while explicitly preventing unsafe actions, ensuring that the agent can aggressively optimize performance without compromising the safety of the robot. We propose a novel method to train the visual locomotion policy end-to-end from pixels using a sample-efficient off-policy RL algorithm. Building upon recent advancements in accelerating RL using demonstrations from previous controllers [21, 22], our algorithm not only learns from its own experiences but also leverages experience generated by the privileged policy. This approach enables the RL agent to rapidly acquire agile skills and adapt behaviors from the privileged experience to suit the specific limitations of its visual sensors, bypassing the computational burden associated with RL from pixels while still ensuring the development of agile and safe legged locomotion skills.

We demonstrate the effectiveness of our approach in simulation. We then directly deploy the learned policy on a real Solo-12 robot and demonstrate the effective acquisition of parkour skills such as crawling, leaping, and climbing obstacles from depth image (see Figure 1). Our approach pushes the robot to its limits: it manages to clear obstacles 1.5 times higher than its height despite the robot being significantly less powerful than the ones typically used in parkour experiments.

In summary, our contributions are as follows:

- we cast parkour learning from depth images as a constrained RL problem,
- we introduce a computationally efficient RL algorithm to train end-to-end visual locomotion policies with significant improvement over methods based on distillation
- and we validate the effectiveness of our approach in simulation and on a real Solo-12 robot to perform parkour skills, outperforming the best movements ever generated with this robot

and reaching performances comparable to recent parkour achievements despite a more limited actuation range.

## 2 Related Work

**Agile Locomotion** RL has demonstrated tremendous success in obtaining robust and adaptive controllers for legged robot [23, 3, 24, 5, 25]. This includes agile skills such as high-speed running [6, 11, 26, 8], recovering from falling [27, 28, 29, 9], jumping [2, 4, 7, 22, 30, 31, 32, 33], climbing obstacles [1, 34, 35, 36, 37, 16, 38, 39, 20, 40, 14, 12, 13] bipedal walking with quadruped robots [41, 30, 22, 13, 42] and walking inside confined spaces [43, 44, 40]. Learning multi-skill locomotion policies, as required in parkour, can be done by training separate policies for each skill, then coordinating them with a high-level planner [28, 45, 46, 14] or distilling them into a single policy [11, 12, 42]. Instead, we follow [13] and learn multi-task policies directly through RL.

**Safe Locomotion** Safety mechanisms have been implemented to ensure safer outcome while performing agile skills [47, 8]. Following [18, 19, 20], we employ constraints alongside rewards in RL to deter undesired behaviors. This not only allows for aggressive optimization of agility while ensuring safety but also simplifies the process of reward tuning for RL. In particular, we exploit the reformulation of Constraints as Terminations [20] (CaT), which was demonstrated as an overhead on top of on-policy RL algorithms [48], and which we extend to the off-policy formulation, more suitable to our case as explained below.

**Sample-Efficient Learning** [49, 22] accelerate RL with demonstrations obtained from trajectory optimization. Other works aimed at exploring RL methods sample efficient enough to train locomotion policies directly in the real-world [29, 50]. Our approach repurposes many of these designs to bypass the computational cost of RL from pixels while obtaining a pure end-to-end RL method, unlocking several advantages exhibited below.

**Vision-Based Locomotion** Prior methods often separate perception from control using intermediate representations such as elevation maps [51, 52, 53, 54, 55, 5, 35, 56, 14, 20], traceability maps [57, 58] or visual odometry [59, 60, 61, 62], for downstream planning and control [63, 64, 65, 66, 67, 68]. Recently, locomotion from pixels [69, 4, 39, 16, 12, 13] has emerged as a powerful paradigm that more tightly coordinates vision and control, often relying on teacher-student approaches (typically by distilling an observation-privileged policy), yet raising some limited action-perception behaviors.

## 3 Method

### 3.1 Agile and Safe Parkour Learning Problem Formulation

Our goal is to train a parkour policy in simulation using RL and transfer it to a real Solo-12 quadruped robot. To this end, we consider an infinite, discounted, constrained Markov Decision Process $(\mathcal{S}, \mathcal{A}, r, \gamma, \mathcal{T}, c_{i \in I})$ with state space $\mathcal{S}$, action space $\mathcal{A}$, reward function $r : \mathcal{S} \times \mathcal{A} \to \mathcal{R}$, discount factor $\gamma$, dynamics $\mathcal{T} : \mathcal{S} \times \mathcal{A} \to \mathcal{S}$ and constraints $\{c_i : \mathcal{S} \times \mathcal{A} \to \mathcal{R}, i \in I\}$. Constrained RL aims to find a policy $\pi : \mathcal{S} \to \mathcal{A}$ that maximizes the discounted sum of future rewards:

$$\max_{\pi} \mathbb{E}_{\tau \sim \pi, \mathcal{T}} \left[ \sum_{t=0}^{\infty} \gamma^t r(s_t, a_t) \right], \tag{1}$$

while satisfying the constraints $c_{i \in I}$ under the state-action policy visitation distribution:

$$\mathbb{P}_{(s,a) \sim \rho_\gamma^\pi, \mathcal{T}} \left[ c_i(s, a) > 0 \right] \leq \epsilon_i \ \forall i \in I, \tag{2}$$

where any value of $c_i(s, a)$ above 0 corresponds to the magnitude of the violation of the $i$-th constraint by taking action $a$ in state $s$.

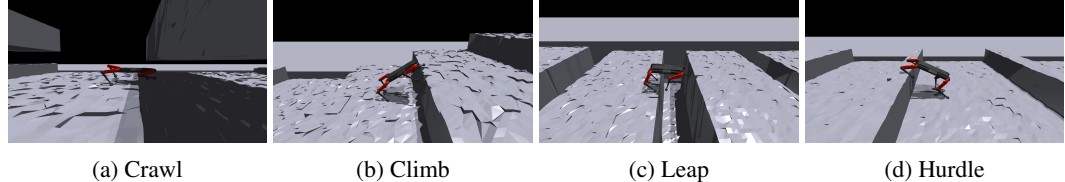

| (a) Crawl | (b) Climb | (c) Leap | (d) Hurdle |

Figure 2: Terrains used to train SoloParkour in simulation: the crawl parkour contains floating objects the robot must crawl under, the step and hurdle parkour contain obstacles for the robot to climb up and down, and the leap parkour contains gaps over which the robot must leap.

**States and Actions** The state $s_t$ corresponds to a history of proprioceptive measurements of the positions and velocities of all 12 joints of the robot and of the orientation and angular velocity of the base of the robot, of previous action samples, a command vector $v^{\text{cmd}}$ in the x-y space given by the user, and of depth images $I^{\text{depth}}_{0,\dots,t}$. The action $a_t$ corresponds to desired joint position offsets with respect to a default joint configuration, that are then converted to torques through a proportional-derivative (PD) controller operating at a higher frequency than the neural policy.

**Rewards** We use a similar reward function to [13] that measures progress towards a direction below a commanded velocity threshold (yet without the additional burden of defining subgoals):

$$r = \min(\langle v, \frac{v^{\text{cmd}}}{\|v^{\text{cmd}}\|_2}\rangle, \|v^{\text{cmd}}\|_2), \tag{3}$$

where $v$ is the velocity of the base relatively in the robot frame. We found that this design was simple while being sufficient for the emergence of agile locomotion skills.

**Constraints** Greedily maximizing Eq. 3 leads to obvious impractical behaviors, in particular exceeding the real robot capabilities and impossible to sim-to-real transfer. To achieve safe and transferable behavior, we enforce a set of constraints $c_i$ that the robot should adhere to. We use straightforward constraint formulations to limit the torques, accelerations, velocities and positions of the joints, avoid unwanted contacts between the robot and the terrain, encourage the robot to head toward the commanded direction and push a specific walking style. For instance, constraints for the torque on the $k$-th joint and for the orientation of the base along the roll axis can be respectively written as:

$$c_{\text{torque}_k} = |\tau_k| - \tau^{\text{lim}} \text{ and } c_{\text{ori}_{\text{roll}}} = |\text{ori}_{\text{roll}}| - \text{ori}^{\text{lim}}_{\text{roll}}, \tag{4}$$

where $\tau^{\text{lim}}$ and $\text{ori}^{\text{lim}}_{\text{roll}}$ are limits that the robot should avoid exceeding. The detailed formulations of constraints are provided Appendix A. We found that the constrained RL formulation, as opposed to reward penalties usually done in RL for locomotion, was crucial for the effective and safe transfer of agile locomotion skills on our real Solo-12 robot while being easier to tune.

**Terrains** The policy is trained to traverse diverse challenging terrains in the IsaacGym simulator [70] to learn a variety of agile locomotion skills, as illustrated in Figure 2. The focus of this work being on parkour, we consider learning locomotion skills such as walking, climbing, leaping, and crawling with four terrain types illustrated in Figure 2. A single policy is jointly trained on a curriculum of variations of these terrains with increasing difficulty [5].

### 3.2 Visual Policy Learning

Ideally, we would like to train the policy from depth pixels to actions using deep RL in order to learn behaviors that best use the limited sensory inputs from the depth cameras. However, directly training the vision policy end-to-end from scratch with RL is impractical, as depth images are costly to render in simulation. An alternative approach is to first train a policy that has access to cheap-to-compute information only accessible in simulation about the terrain surrounding the robot, then distill the behaviors learned from this privileged information into a visual policy observing a history of depth images using imitation learning [16, 13, 12]. While the experimental setup can be chosen

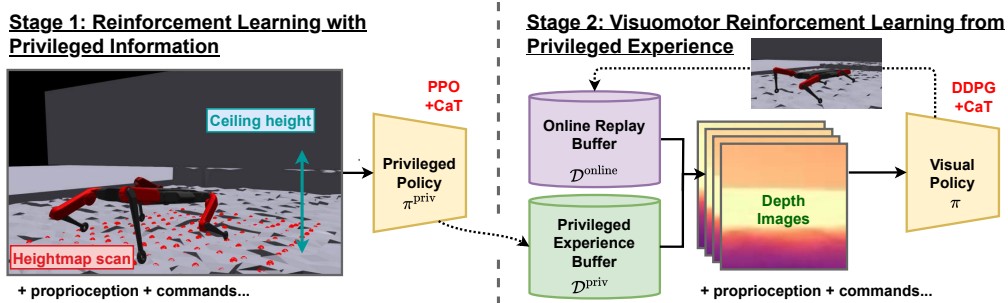

**Stage 1: Reinforcement Learning with Privileged Information**

**Stage 2: Visuomotor Reinforcement Learning from Privileged Experience**

Figure 3: SoloParkour leverages a two-stage RL approach to train visual locomotion policies in simulation. Stage 1: we train a privileged policy that observes a heightmap scan of its surroundings and the height of the nearby floating objects using PPO with Constraints as Terminations (CaT) [20]. Stage 2: we train a policy from depth pixels using a variant of DDPG with CaT that learns from a dataset of privileged experience collected using the Stage 1 policy.

to reduce the distillation gap (e.g. by choosing specific sensors providing extensive observability), there is in general no guarantee that the learned behaviors will transfer well to the visual policy due to the observability gap between the privileged information and the visual sensors. For instance, elevation maps may contain information about surfaces that are not visible from the camera inputs. We then propose a generic solution mitigating these two problems.

Our approach, SoloParkour, is illustrated in Figure 3. Following prior works [16, 12, 13], we first learn a privileged policy that has access to cheap-to-compute information about the terrain surrounding the robot. Unlike [16, 12, 13], we propose to use this privileged policy to warm-start the reinforcement learning of an end-to-end visual policy. This way, we let the RL agent decide on the best behaviors given its limited sensory inputs while bypassing the computational complexity of training from depth images.

**Stage 1: Privileged Policy Learning** We first train a privileged policy $\pi^{\mathrm{priv}}$ that has access to cheap-to-compute information about the terrain surrounding the robot. In place of histories of depth images and proprioception, the robot observes the privileged state $s_t^{\mathrm{priv}}$ that includes the current proprioceptive measures, velocity commands, and the previous action as well as a height-scan map of its surrounding and the ceiling height. In crawl terrains, the ceiling height corresponds to the distance from the ground to a levitating block when such a block is near the robot. In other terrains, or when no levitating blocks are nearby, the ceiling height is set to an arbitrarily high default value. We found that this provides sufficient information about the terrain geometry to train the privileged policy for all the proposed terrain tracks.

The privileged policy is parameterized by a Multi-Layer Perceptron (MLP) trained using Proximal Policy Optimization [48] (PPO), an on-policy RL method widely used in learning-based locomotion. We use CaT [20] to enforce the constraints.

**Stage 2: End-to-End Vision-Based RL from Privileged Experience** To overcome the computational complexity of RL from pixels, we propose to adapt Reinforcement Learning with Prior Data [21] (RLPD) to transfer the experience from the privileged policy to visual locomotion and learn from depth images in a sample efficient manner. We build on design principles from [29, 21, 22] and implement off-policy RL with a high update-to-data ratio. Compared to on-policy approaches such as PPO, this minimizes the number of depth image rendering steps in simulation in favor of increased updates from the privileged and online experience.

More precisely, we employ a variant of Deep Deterministic Policy Gradient [71] (DDPG) that uses two replay buffers: its online replay buffer $\mathcal{D}^{\mathrm{online}}$ and a privileged experience buffer $\mathcal{D}^{\mathrm{priv}}$. $\mathcal{D}^{\mathrm{priv}}$ is constructed by employing the fully trained $\pi^{\mathrm{priv}}$ from Stage 1 to generate demonstrations that include the depth modality $I^{\mathrm{depth}}$. During policy learning, batches of experiences are sampled in equal proportion from $\mathcal{D}^{\mathrm{online}}$ and $\mathcal{D}^{\mathrm{priv}}$ to train the policy $\pi$ and its Q-function. We use REDQ [72] and

Layer Normalization [73] to stabilize Q-learning at high update-to-data ratio [74, 21, 22]. Importantly, while the visual actor $\pi$ is trained end-to-end from a history of depth images, we found that training the critic from privileged state $s_t^{\mathrm{priv}}$ rather than on the full state $s_t$ in an asymmetric actor-critic fashion [75] was faster and more stable. We incorporate CaT [20] in an off-policy manner to learn visual locomotion that keeps satisfying the constraints.

The resulting RL algorithm is sample efficient enough to train our visual policies, parameterized by a ConvNet [76] to process depth images individually, a Gated Recurrent Unit [77] to handle histories of observations, and an MLP head, with RL directly from pixels in simulation.

## 4 Experiments

### 4.1 Experimental setup

**Simulation and robot**    The policies are trained in the IsaacGym [70] simulator using massively parallel environments. The full policy learning pipeline can be trained on a single NVIDIA RTX 4090 GPU in less than 20 hours. After training in simulation, the controller is directly deployed on a real Solo-12 quadruped robot. The policy runs at $50Hz$ on a Raspberry Pi 5. Target joint positions are sent to the onboard PD controller running at $10kHz$. We use an Intel RealSense D-405 stereo camera to capture depth images and process them to resolution $48 \times 48$. While the depth images are rendered every 5 environment steps in simulation (i.e. 10Hz in the time reference of the simulation), we provide the images at the speed of the depth pipeline on the real robot at $30Hz$.

When standing, the height of Solo is 26cm and its body length is 45cm, which is similar to the Unitree Go-1 used in [12, 13]. However, Go-1 has a thrust-to-weight ratio 2 to 3 times superior to Solo. Thus, we don't expect to overcome obstacles as challenging as [12, 13].

**Baselines and ablations**    We validate our approach in simulation and compare SoloParkour to the following baselines and ablations.
- DAgger [17]: the method used in [13, 12, 16] to distill the privileged policy into the visual policy using imitation learning through action relabelling with the privileged policy.
- Behavior Cloning (BC): distilling the privileged policy by training the visual policy directly with supervised learning on the privileged experience $\mathcal{D}_{\mathrm{priv}}$.
- Privileged Reconstruction: training the vision module to reconstruct the privileged information from the history of depth and proprioceptive inputs, then reemploy the Stage 1 policy based on these reconstructions, aimed to resemble [16, 56].
- From Scratch: an ablation of our approach where we train the visual policy with RL from scratch, without privileged experience $\mathcal{D}_{\mathrm{priv}}$.
- No Priv. Critic: an ablation of our approach where the critic of Stage 2 observes histories of depth images instead of the simplified privileged information.
- Visual RL w/o CaT: an ablation of our approach where we train the visual RL policy without constraints (but from experience from the same constrained Stage 1 policy).

We first train a privileged policy from Stage 1. Then, except for BC which is much faster to train as it doesn't require querying the simulator, we train all these baselines and ablations with the same computational budget. Finally, we compare their performance by executing the learned policies in simulation and measuring the distance traveled in the terrains of Figure 2.

### 4.2 Simulation Experiments

**Comparison to supervised distillation**    In Figure 4a, we compare SoloParkour against DAgger, BC and *Privileged Reconstruction*. For comparison, we also report the performances of the privileged policy used to train these methods. SoloParkour performs roughly as well as the privileged policy on the hurdle, step, and crawl tracks and marginally worse on the leap track. It outperforms the supervised distillation baselines on all terrains and at almost all levels of difficulty. The difference is particularly significant against DAgger and BC on the leap terrains, where the robot must

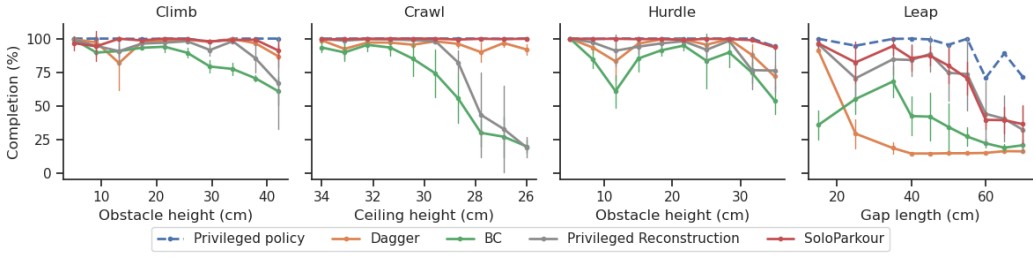

(a) Comparison between SoloParkour and imitation-based distillation methods.

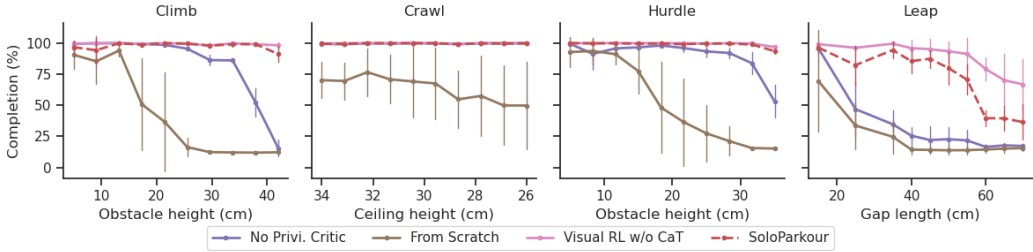

(b) Ablations of the proposed methods.

Figure 4: Average terrain completion (over 4 training seeds) by obstacle dimension for each policy.

perform a complex motion to overcome the gaps, requiring perfect timing to succeed. On these terrains, DAgger and BC often fail to propel themselves enough and miss the edge of the landing side of the platform, or ignore the gap and fall into it. Meanwhile, *Privileged Reconstruction* struggles on the crawl and climb terrains, where occlusions hinder accurate reconstruction of the surrounding terrain geometry. We attribute these differences to our end-to-end training from pixels with the RL objective, which results in tighter coordination between vision and control, understood as a local action-perception refinement.

**Ablative analysis**  In Figure 4b, we examine the importance of two design choices for SoloParkour given the same computation budget. Training the visual policy from scratch without privileged experience (*from Scratch*) is much less sample-efficient and achieves poor performances overall, validating the importance of transferring privileged experience from the Stage 1 policy. Having the critics process the full-depth observation instead of only the privileged information (*No Priv. Critic*) causes additional computation overheads during Q-learning, which are yet unnecessary to train good visual policies in our setting. As a result, given the same computation budget, *No Priv. Critic* processes fewer samples and achieves lower performances than the full SoloParkour method.

**Constraints satisfaction**  In Figure 5, we examine the violation of the torque constraint for the front left shoulder joint and the base orientation constraint, as outlined in (4). We focused exclusively on successful trajectories where the robot managed to navigate the entire level in order to exclude data from extreme states, such as when the robot is falling outside of the track or gets endlessly stuck on a high obstacle. Hence, we report results for only two gap lengths on the leap track for DAgger, as it fails to overcome larger gaps. Thanks to

| Step | Leap | Crawl | Average |
|------|------|-------|---------|
| 85%  | 70%  | 100%  | 85%     |

Table 1: Success rates achieved by the policies for different obstacles on the real robot, averaged over 2 training seeds and 10 trials per obstacle per seed.

off-policy CaT, SoloParkour maintains constraint compliance effectively after visuomotor RL, with torque violations under 4% and base orientation violations under 10% on most levels, despite the highly dynamic skills involved to overcome the obstacles. These results are comparable to those of the privileged policy and supervised distillation baselines. By contrast, *Visual RL w/o CaT* achieves higher terrain completion rates but at the cost of significantly increased constraint violations, rendering the policies unsuitable for real-world robot transfer. Additional results on the satisfaction of a broader set of constraints are discussed in Appendix C.

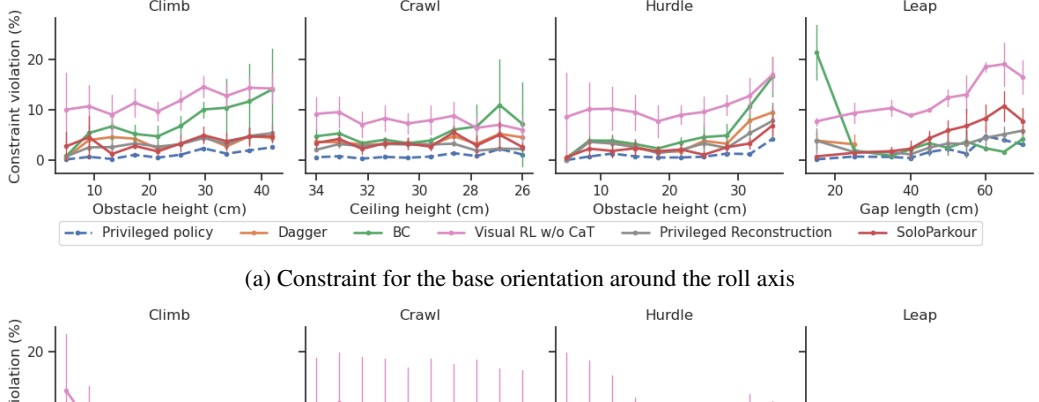

(a) Constraint for the base orientation around the roll axis

(b) Torque constraint for the front left knee joint

Figure 5: Constraint violations (in %) when the policies successfully traverse the terrain level by obstacle dimension, averaged across 4 training seeds.

### 4.3 Real-Robot Experiments

We deploy SoloParkour directly on the real Solo-12 and evaluate it to traverse challenging obstacles that involve climbing, crawling, and leaping (see Figure 1). We found that the policy handled the depth vision signals sent by the camera, responding synchronously to obstacles at the right time. Table 1 reports the results achieved by SoloParkour. Our approach successfully climbs a 40cm step (1.5 times the robot's height), leaps over a 35cm gap (78% of its length), and crawls under obstacles 20cm above the ground with high repeatability.

## 5    Limitations

The experiments on the real hardware have revealed that our policy, while respecting the motor limitations, is likely hitting the battery current limits. This issue was not anticipated, hence not modeled neither in the simulation nor in the imposed constraints. It likely was amplified during the course of the experimental study by damaging the batteries when exceeding their capabilities. An exciting direction is to properly model this physical effect in the constrained-MDP, which has the potential to improve the performances of all other robots in future parkour research.

Moreover, current parkour learning approaches, including ours, require that simulation terrains be manually constructed for each specific skill. Consequently, robots can only learn new skills by designing new types of terrain. Future work could explore procedural or generative simulators to create more diverse and realistic environments for training locomotion policies and transition from depth-based to RGB vision for legged robots.

## 6    Conclusion

We introduced SoloParkour, a novel method that combines constraints and sample-efficient RL from privileged experience to train agile and safe locomotion for legged robots end-to-end from depth pixels. We found that constraints simplify the work of designing the MDP while leading to more consistent results. The end-to-end resolution leads to better training convergence compared to previous teacher-student approaches.We also experimentally brought the lightweight robot Solo-12 closer to its limit than ever before, achieving various parkour stages despite severe actuator limitations.

## Acknowledgements

This work was funded in part by ANITI (ANR-19-P3IA-0004), COCOPIL (Région Occitanie, France), PEP# O2R (AS2 ANR-22-EXOD-0006), Dynamograde (ANR-21-LCV3-0002), ROBO-TEX 2.0 (ROBOTEX ANR-10-EQPX-44-01 and TIRREX-ANR-21-ESRE-0015) and the National Science Foundation (grants 2026479, 2222815 and 2315396). It was granted access to the HPC resources of IDRIS under the allocations AD011012947 and AD011015316 made by GENCI.

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

# A Implementation Details

## A.1 Rewards and Constraints

We use the reward function 3 from [13] that measures progress toward a specific direction. To always have positive rewards, we add a survival bonus of $0.5$ at each time step, and, following [5], we clip total rewards below $0.0$.

To enforce the constraints, we follow CaT [20] and reformulate the constrained RL problem 1 into the following RL problem:

$$\max_{\pi} \mathbb{E}_{\tau \sim \pi} \left[ \sum_{t=0}^{\infty} \left( \prod_{t'=0}^{t} \gamma(1 - \delta(s_{t'}, a_{t'})) \right) r(s_t, a_t) \right],$$ (5)

with termination probabilities $\delta(s_t, a_t)$. Following CaT, we define the termination probabilities as:

$$\delta = \max_{i \in I} p_i^{\max} \text{clip}(\frac{c_i^+}{c_i^{\max}}, 0, 1),$$ (6)

where $c_i^+ = \max(0, c_i(s, a))$ is the violation of constraint $i$, $c_i^{\max}$ is an exponential moving average of the maximum constraint violation over the last batch of experience collected in the environment, and $p_i^{\max}$ a hyperparameter that controls the maximum termination probability for the constraint $i$.

Table 2 lists all the constraints used. Following [20], we separate constraints between hard constraints, where $p_i^{\max} = 1.0$, and soft constraints, where $p_i^{\max}$ increases throughout the course of training, allowing the RL agent to discover agile locomotion during the early stage of training while enforcing more the constraints later on to ensure safe behaviors. To encourage the emergence of a specific locomotion style, some constraints are activated only in specific settings. For instance, the *Stand still* constraints $c_{\text{still}}$ are only active when no velocity command is provided, whereas the *Base orientation* and *Number of foot contacts* are only active on flat terrains and on early terrain levels.

We found that rescaling the constraint violations by the square root function $c_i \leftarrow \sqrt{c_i^+}$ helps CaT be less sensitive to extreme values of constraint violations.

| Type | Expression | Hard | Cond. |
|---|---|:---:|:---:|
| Knee or base collision | $c_{\text{knee/base contact}} = 1_{\text{knee/base contact}}$ | ✓ | ✗ |
| Foot contact force | $c_{\text{foot contact}_j} = \|f^{\text{foot}_j}\|_2 - f^{\text{lim}}$ | ✓ | ✗ |
| Foot stumble | $c_{\text{stumble}_j} = \|f_{\text{xy}}^{\text{foot}_j}\|_2 - 4\|f_{\text{z}}^{\text{foot}_j}\|$ | ✗ | ✗ |
| Heading | $c_{\text{heading}} = \|\text{angle}_{\text{base}} - \text{angle}_{\text{cmd}}\| - \text{angle}^{\text{lim}}$ | ✗ | ✗ |
| Torque | $c_{\text{torque}_k} = \|\tau_k\| - \tau^{\text{lim}}$ | ✗ | ✗ |
| Joint velocity | $c_{\text{joint velocity}_k} = \|\dot{q}_k\| - \dot{q}^{\text{lim}}$ | ✗ | ✗ |
| Joint acceleration | $c_{\text{joint acceleration}_k} = \|\ddot{q}_k\| - \ddot{q}^{\text{lim}}$ | ✗ | ✗ |
| Action rate | $c_{\text{action rate}_k} = \frac{\|\Delta q_{t,k}^{\text{des}} - \Delta q_{t-1,k}^{\text{des}}\|}{dt} - \dot{q}^{\text{des lim}}$ | ✗ | ✗ |
| Joint limits min | $c_{\text{joint}_j^{\min}} = \text{joint}_j^{\min} - \text{joint}_j$ | ✗ | ✗ |
| Joint limits max | $c_{\text{joint}_j^{\max}} = \text{joint}_j - \text{joint}_j^{\max}$ | ✗ | ✗ |
| Foot air time | $c_{\text{air time}_j} = t_{\text{air time}}^{\text{des}} - t_{\text{air time}_j}$ | ✗ | ✗ |
| Base orientation (roll-axis) | $c_{\text{ori}_{\text{roll}}} = \|\text{ori}_{\text{roll}}\| - \text{ori}_{\text{roll}}^{\text{lim}}$ | ✗ | ✗ |
| Base orientation | $c_{\text{ori}} = \|\text{base ori}_{\text{xy}}\|_2 - \text{base}^{\text{lim}}$ | ✗ | ✓ |
| Number of foot contacts | $c_{\text{n foot contacts}} = \|n_{\text{foot contact}} - n_{\text{foot contact}}^{\text{des}}\|$ | ✗ | ✓ |
| Stand still | $c_{\text{still}} = \|q - q^{\star}\|_2 - \epsilon_{\text{still}}$ | ✗ | ✓ |

Table 2: List of constraints, where *Hard* indicates whether each row corresponds to a hard constraint and *Cond.* indicates whether a constraint is active only under certain conditions.

## A.2 Policy Learning

We built our RL algorithm with the CleanRL [78] implementations of PPO and DDPG. During privileged policy learning, we linearly increase the damping parameter (Kd) of the PD controller from $0.05$ to $0.2$ but keep it fixed at $0.2$ during visual policy learning. Indeed, we empirically observed that RL discovers agile skills more easily with lower Kd but policies with higher Kd transfer better to the real Solo-12.

**Privileged Policy Learning**    An MLP parametrizes the privileged policy with hidden dimensions $[512, 256, 128]$ and elu activations. We use PPO [48] with 4096 actors in parallel in simulation. The training procedure is very similar to [5, 13, 20].

**Visual Policy Learning**    The actor processes depth images using a vision neural network consisting of three blocks of a convolution with leaky ReLU activations, followed by max pooling and a linear layer to produce the depth embeddings. Random translation, random noise, and random cutout are applied to the depth images during training. The actor then processes the history of proprioceptive information, actions, and depth embeddings with a one-layer Gated Recurrent Unit [77] (GRU) of hidden size 256. This GRU is followed by a MLP with hidden dimensions $[512, 256, 128]$ and elu activations. The output of the final layer is processed by a tanh activation function and rescaled to produce the 12-dimensional action vector. We used the action bounds observed in the privileged experience buffer to rescale the actions given by the actor.

The critic network is parameterized by a MLP with hidden dimensions $[512, 256, 128]$, layer normalization and elu activations. While the actor observes the full state $s_t$, which includes a history of depth images, the critics process the privileged state $s_t^{\text{priv}}$, which includes privileged heightmap scan and ceilings instead of high-dimensional images.

We generate trajectories from the privileged policy that amounts to 2 million state-action samples and store them in the privileged experience buffer $\mathcal{D}^{\text{priv}}$ whereas online experience is collected by 256 actors in parallel into the online replay buffer $\mathcal{D}^{\text{online}}$. We store the constraint violations $c_i^+$ of both online and privileged experience in their respective replay buffers to recompute the termination probabilities $\delta$ on the fly during off-policy learning. Both $\mathcal{D}^{\text{priv}}$ and $\mathcal{D}^{\text{online}}$ store privileged information at every step and depth image every 5 environment steps. During training, we only give the vision network one image every five timesteps and then replicate the depth latent five times to match the sequence length of the other observations. The online replay buffer stores the GRU hidden latent produced by the online actors whereas the privileged replay buffer stores zeros for these latents. This is done to initialize the first hidden of the DDPG actor correctly during off-policy training.

We train the visual policy using a variant of RLPD [21]. We build upon DDPG [71] with an update-to-data ratio of 8 during policy evaluation. We use REDQ [72] with 10 critics and an ensemble of 2 random critic targets.

## A.3 Baselines

The BC baseline uses the same neural network architecture as SoloParkour, as described in Section A.2. We train the BC policies to regress the action based on the history of observations on the same dataset of demonstrations $\mathcal{D}^{\text{priv}}$ generated by the privileged policy $\pi^{\text{priv}}$ as SoloParkour.

The DAgger baseline uses the same neural network architecture as SoloParkour and BC. We employ the Stage 1 policy $\pi^{\text{priv}}$ as teacher for action relabelling. We found that starting the DAgger policy from the BC-pretrained weights greatly improves online learning efficiency.

For the *Privileged Reconstruction* baseline, we use the same observation space and neural architecture as SoloParkour, BC and DAgger for the vision module except that the output of the GRU is projected to the privileged information space through a linear layer. We employ the privileged policy $\pi^{\text{priv}}$ as frozen MLP head and only train the vision module to reconstruct the privileged information from the history of past observations and actions. Similar to [16] and unlike [56], we found it highly

beneficial to train the vision network with experience generated by the resulting online policy, rather than training solely on the fixed dataset $D^{\text{priv}}$ generated by the privileged policy.

The *No Priv. Critic* ablation processes visual input in the critic network instead of privileged information. Its vision module follows the same architecture as the visual policy.

For the *Visual RL w/o CaT* ablation, we use the same dataset of privileged experience $\mathcal{D}^{\text{priv}}$ as the one used to train all other methods (except for the *From Scratch* ablation that doesn't learn from any demonstration), but we remove the constraints for Stage 2 RL. Note that the privileged experience $\mathcal{D}^{\text{priv}}$ was generated with $\pi^{\text{priv}}$ which was trained with Constrained RL and therefore satisfies the constraints.

## B   Real Robot Setup

We use the Solo-12 quadruped robot for our experiments. We built a custom 3D-printed plastic piece to mount the Intel RealSense D405 stereo camera observing in front of the robot. We use the Python wrapper of librealsense to capture depth images at resolution $424 \times 240$. We resize and crop the images to $48 \times 48$ and apply the librealsense postprocessing hole-filling filter. Depth images are preprocessed in a separate thread on a separate CPU as they come, at around 30Hz. The visual policy runs at 50Hz using ONNX and produces target joint angles to torque by a PD controller with stiffness $Kp = 4.0$ and damping $Kd = 0.2$ running at 10KHz. Hence, depth embeddings are updated at a higher frequency at inference than during training. All the computation is done through Python scripts by the onboard Raspberry Pi 5. Velocity commands are sent to the embedded controller via a wireless gamepad.

## C   Additional Results

In Figure 6 and 7, we present further results on constraint satisfaction for SoloParkour and the baselines introduced in Section 4.1. SoloParkour demonstrates high constraint satisfaction, highlighting the effectiveness of our approach in achieving safe yet agile locomotion skills over challenging terrains using vision.

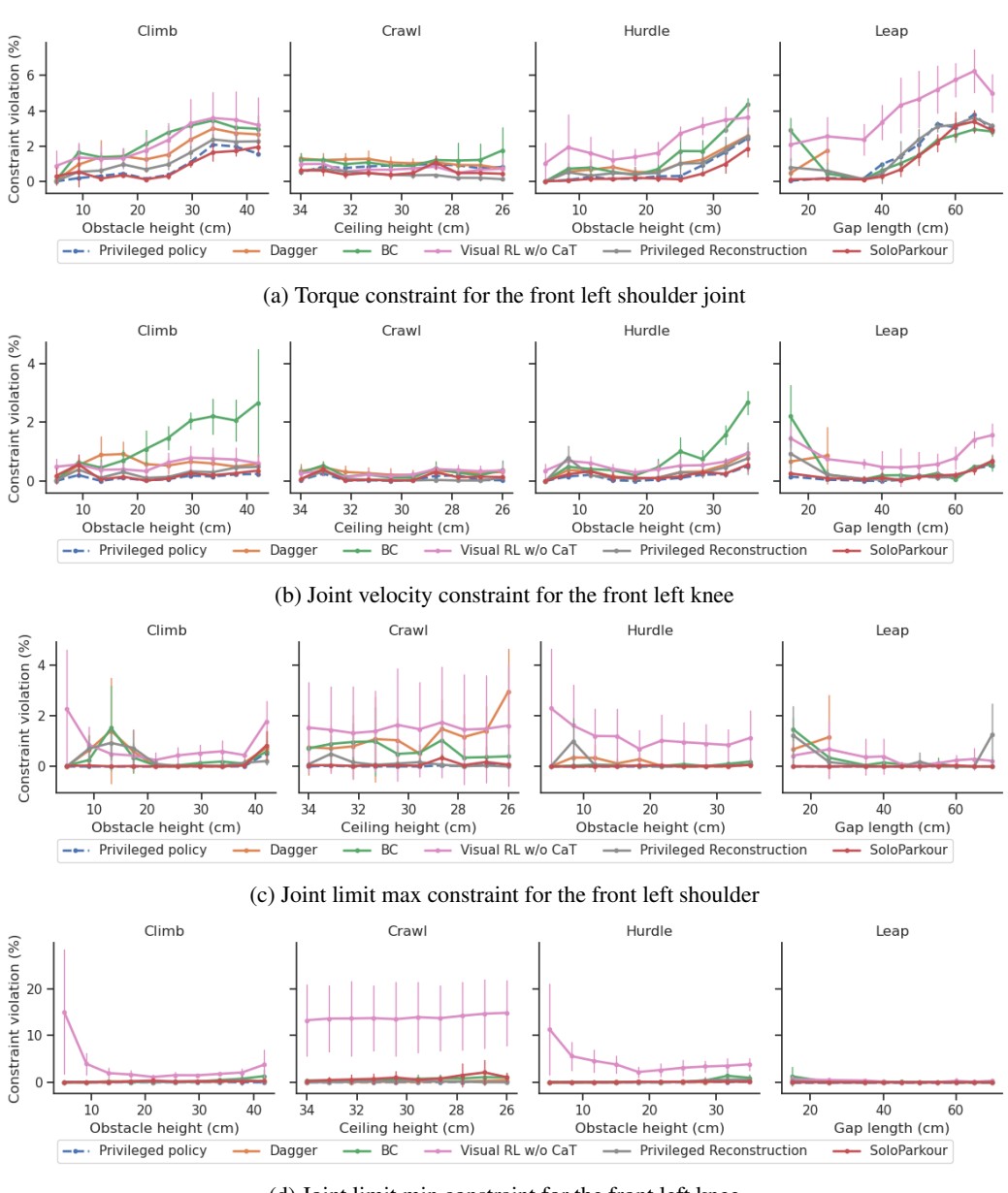

(a) Torque constraint for the front left shoulder joint

(b) Joint velocity constraint for the front left knee

(c) Joint limit max constraint for the front left shoulder

(d) Joint limit min constraint for the front left knee

Figure 6: Constraint violations (in %) when the policies successfully traverse the terrain level by obstacle dimension, averaged across 4 training seeds.

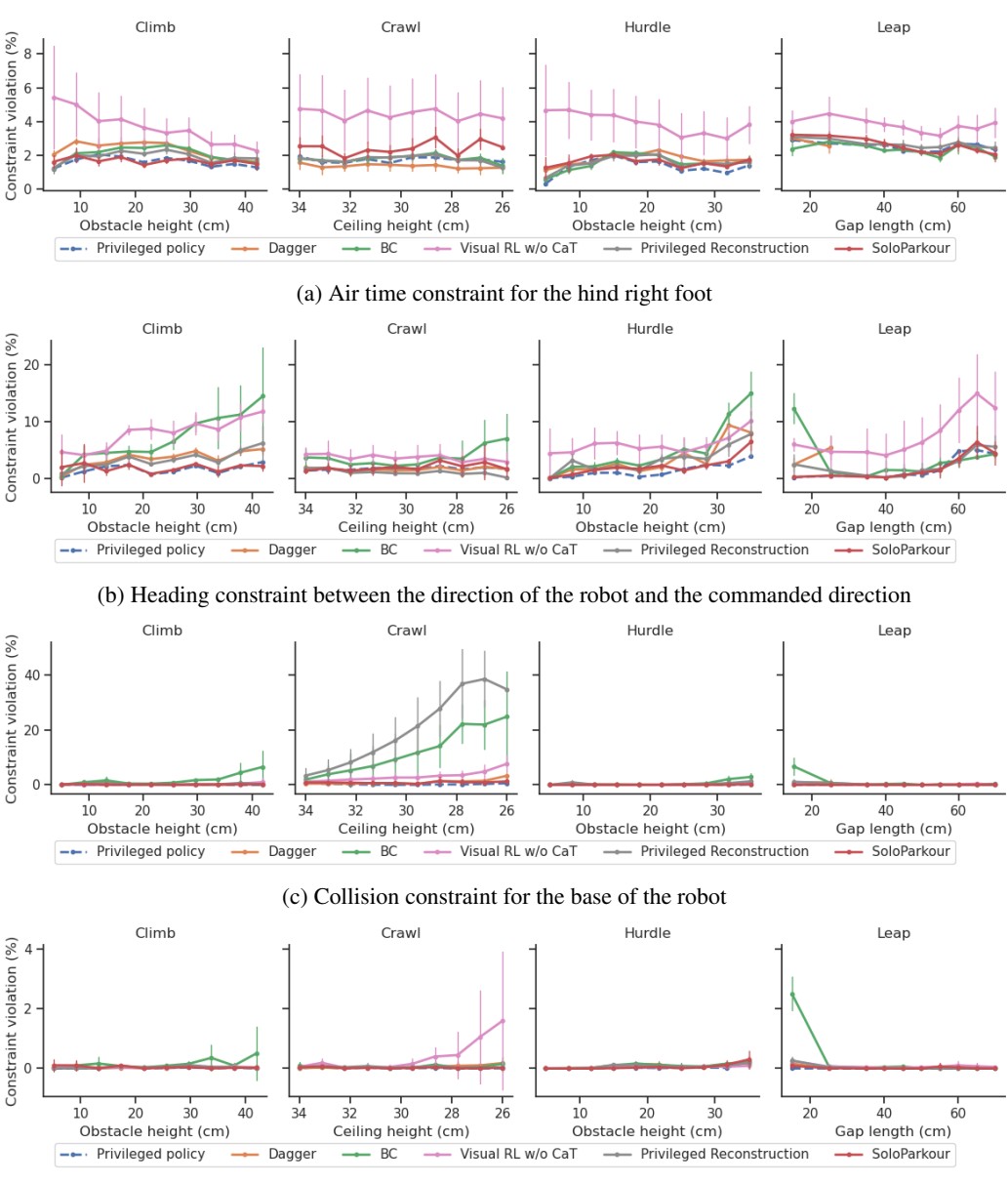

(a) Air time constraint for the hind right foot

(b) Heading constraint between the direction of the robot and the commanded direction

(c) Collision constraint for the base of the robot

(d) Collision constraint for the hind left knee

Figure 7: Constraint violations (in %) when the policies successfully traverse the terrain level by obstacle dimension, averaged across 4 training seeds.

