# OpenReview forum: "SoloParkour: Constrained Reinforcement Learning for Visual Locomotion from Privileged Experience"
_robot-learning.org/CoRL/2024/Conference — CoRL 2024_

### Official Review · Reviewer_1Wye · 2024-07-11
**Valuable insight but lack of convincing experiments**

**Originality:** 3
**Technical Quality:** 3
**Clarity Of Presentation:** 5
**Potential Impact:** 3
**Recommendation:** 2
**Confidence:** 5

**Review:**

In general, this paper is clear and well-written. From the writing perspective, this paper is well-organized and clearly describes the motivation, proposal, and experiment implementation. The proposal to restrict the torque constraint violation in the parkour task is important and valuable. However, the experiments and the major constraints this work faces are not convincing enough to show the difference between the proposed method and the baseline. Some trivial solutions to bound the torque constraint violation are not considered in the baseline, which makes the comparison unfair.

The following are the strengths and weaknesses of this work:

**strength**
* Valuable research on constraint violation of the deployed policy.

* Valuable insight on using DDPG mixed with expert trajectory.

**weakness**
* Lack of **Hurdle** experiments in the real world.

* The experiments are not convincing enough to show the effectiveness of reducing the torque constraint violation. Using an external power source to push the system to its limit and show the effectiveness of the proposed method would be more convincing.

**Quality Of The Limitations Section:**

2

**Questions For Rebuttal:**

1. Since the author trains the deployable policy $\pi$ based on a privileged policy $\pi^\text{priv}$ from PPO, does the deployable policy reuse any weights of the privileged policy $\pi^\text{priv}$?

2. What is the linear velocity of the robot during the parkour course?

3. Is it possible to directly set the hardware constraint in the PD controller? Does it help with the torque constraint in the real-world robot?

4. What are the major causes of failures in the leaping task?

**Robotics Focus:**

4

**Summary Of Paper:**

This work proposes employing constraint as termination in training the extreme challenging parkour tasks for quadurped robot. Using DDPG with RLPD, the author trains a visual parkour policy that reduces the chances of violating the torque constraint in the parkour tasks.

**Summary Of Recommendation:**

This paper provides valuable insight on the constraint violation of the deployed policy. However, the experiments are not convincing enough to show the effectiveness of reducing the torque constraint violation. I recommend a weak reject on this paper.

---

### Official Review · Reviewer_uDaM · 2024-07-17
**This paper presents a novel system for training visual locomotion on a quadrupedal robot using constrained reinforcement learning and off-policy distillation techniques.**

**Originality:** 2
**Technical Quality:** 3
**Clarity Of Presentation:** 3
**Potential Impact:** 3
**Recommendation:** 4
**Confidence:** 5

**Review:**

**Strengths**

- Demonstrates a complete system for training visual locomotion over varied terrains on a real quadrupedal robot.
- Introduces incremental methods to enhance state-of-the-art techniques, including "soft" constraints-based termination in RL and sample-based distillation for policy/Q-function learning using a teacher policy learned with privileged information in simulation.
- Provides simulation-based comparisons with other methods such as Dagger, BC, and training from scratch for vision-based locomotion behaviors.

**Weaknesses**

- **Comparison with Other Baselines:** The paper lacks comparisons with methods that distill depth/point cloud into visual features, either as implicit feature vectors or explicit local heightmap neural networks.
- **Effects of CaT:** The impact of CaT on visual locomotion is unclear. The paper needs more comprehensive data and analysis to determine its usefulness in the domain of visual locomotion.
- **Missing Details:** The manuscript lacks clarity on the inputs for the privileged critic, the rationale for choosing privileged observations, and the robot's behavior in the hardware video after navigating ceiling-type terrain.

**Quality Of The Limitations Section:**

2

**Questions For Rebuttal:**

**1. Comparison of Other Baselines:** The paper proposes an incremental improvement for policy distillation on visual locomotion tasks, with comparisons to action distillation methods like Dagger and BC shown in Figure 4a. However, it would be beneficial to include the folllowing references and compare the following methods, which distill depth/point cloud into visual features either as an implicit feature vector (partially shown in [1]) or as an explicit local heightmap neural-network-based method [2, 3], and then feed them into the privileged policy for online execution. This approach avoids potential suboptimal results from action imitation. While hardware comparisons are challenging, simulations could enhance the discussion of a comprehensive set of methods for visual locomotion.

[1] https://vision-locomotion.github.io/

[2] https://arxiv.org/abs/2306.14874

[3] https://arxiv.org/abs/2309.14594

**2. Effects of the CaT:** Although the original CaT is previously published, its overall effects on visual locomotion remain unclear. Constraints like unwanted collisions are critical for achieving agile locomotion over terrains. The paper only presents results on torque limits in Table 1 and Figure 5. A more thorough investigation of CaT's impact on specific visual locomotion tasks is necessary to determine its usefulness in this domain. The data collected for Table 2 and Figure 5 appear insufficient; a larger dataset would provide a better understanding of CaT's effect on learned behavior. Additionally, it is crucial to assess whether CaT's absence during the distillation phase affects the final policy. What if the CaT is removed during distillation?

**3. Missing Important Details:**
- Does the privileged critic only use an additional heightmap compared to a regular critic? If not, what are the inputs? What is the rationale for choosing the privileged observation?
- In the hardware video, why does the robot squat after walking through the ceiling-type terrain? Is it because the GRU overfits to the ceiling, causing the robot to remain crouched afterward?

**Robotics Focus:**

4

**Summary Of Paper:**

Parkour presents a significant challenge for legged robots, requiring agile and precise navigation through complex environments with limited sensory inputs. This paper introduces a novel method for training end-to-end visual policies, transforming depth pixels into robot control commands to achieve agile and safe quadruped locomotion. The approach frames robot parkour as a constrained reinforcement learning (RL) problem, aiming to maximize agile skill emergence within the robot's physical limits while ensuring safety. Initially, a policy is trained without vision using privileged information about the robot's surroundings. This privileged policy generates experience to warm-start a sample-efficient off-policy RL algorithm from depth images, enabling the robot to adapt behaviors to visual locomotion without the high computational costs of RL directly from pixels. The method's effectiveness is demonstrated on a real Solo-12 robot, showcasing its ability to perform various parkour skills such as walking, climbing, leaping, and crawling.

**Summary Of Recommendation:**

While the manuscript demonstrates a comprehensive system for training visual locomotion on a real quadrupedal robot and introduces innovative methods to enhance state-of-the-art techniques, several weaknesses need to be addressed to strengthen the paper. These include the lack of comparison with other baseline methods that use depth/point cloud features, unclear impacts of the CaT on visual locomotion with insufficient data provided, and missing critical details about the privileged critic's inputs and the robot's behavior in the hardware video. Addressing these concerns would significantly enhance the manuscript's contribution to the field and my support for its publication.

---

### Official Review · Reviewer_vns5 · 2024-07-20

**Originality:** 4
**Technical Quality:** 5
**Clarity Of Presentation:** 5
**Potential Impact:** 3
**Recommendation:** 3
**Confidence:** 2

**Review:**

The paper introduces a promising approach to efficiently train the visual-based legged robot in constrained MDP using a privileged policy that has access to surrounding terrain information. The approach is well-explained and experiments are convincing.

The task setting, rewards, and constraints are clear as well as the method description and its motivation.

I like the choice of baselines and ablations. Dagger and Behavior Cloning are common approaches for such tasks. Training from scratch and a critic without privileged information are natural alternatives to the proposed method. The resulting plots are convincing in my opinion.

The limitation section is interesting, the authors notice and highlight the important problems of battery limits as well as requirements to design new types of terrains for training new skills. These problems are important and common in many robotic fields.

The authors also provide sufficient information to reproduce their results.

In my opinion, it is a clean, easy-to-follow paper, with explicit experimental sections and sufficient hardware demonstration.

**Quality Of The Limitations Section:**

3

**Questions For Rebuttal:**

* Typo on line 271 - “MPD”

**Robotics Focus:**

4

**Summary Of Paper:**

This work presents an approach to training agile-legged robot using visual observation. The paper proposes to use a privileged policy, that has access to information about the terrain around the robot at the first stage to generate trajectories. Then in the second stage, the agent is trained with the help of the generated trajectories while using only depth images for policy training and additional privileged information for a critic. The experimental section considers different approaches for such tasks showing the superiority of the method in simulation. The paper also presents experiments on physical robot.

**Summary Of Recommendation:**

The paper is convincing and well-written. Results are extensively demonstrated on hardware.

---

### Author Rebuttal · Authors · 2024-08-09

We thank the reviewers for their insightful comments on our work.
We have accordingly updated the paper and its appendix (paper.pdf), where relevant modifications are colorized in blue.
We notably added new baselines and experimental results in Section IV and Appendix C.

---

### Decision · Program_Chairs · 2024-09-04

**Decision:**

Accept

**Comment:**

The paper demonstrates a learning-based Parkour task on the Solo quadrupedal robot using constrained RL. Overall, the reviewers agreed that the results are convincing and impressive. The paper is also clearly written. However, they also encouraged the authors to include more analysis, particularly comparisons against baselines. Please refer to all the reviewers for more details.

During the rebuttal phase, we got polarized recommendations from the reviewers. One reviewer seemed to be happy about the result and increased the score to SA. On the other hand, another reviewer still remains negative despite a few resolved concerns. As an area chair, I am a bit toward the positive side in the hope that this paper with dense hardware experiments can still be a good reference to future researchers, even when there are somewhat similar papers out there. Therefore, I would like to recommend accepting this manuscript.